# Remote Sensing Change Detection Based on Unsupervised Multi-Attention Slow Feature Analysis

**Weipeng Jing [1,\*], Songyu Zhu [1,2], Peilun Kang [1], Jian Wang [3], Shengjia Cui [4], Guangsheng Chen [1] and Houbing Song [5]**

1 College of Information and Computer Engineering, Northeast Forestry University, Harbin 150040, China; zhusongyu@nefu.edu.cn (S.Z.); 374774222@nefu.edu.cn (P.K.); kjc_chen@nefu.edu.cn (G.C.)
2 College of Electronics and Information Engineering, Harbin Vocational & Technical College, Harbin 150081, China
3 Aerospace Information Research Institute, Beijing 100094, China; wangjian@radi.ac.cn
4 Baidu Company Ltd., Beijing 100085, China; cuishengjia@baidu.com
5 Department of Electrical, Computer, Software and Systems Engineering, Embry-Riddle Aeronautical University, Daytona Beach, FL 32114, USA; songh4@erau.edu
\* Correspondence: jwp@nefu.edu.cn; Tel.: +86-186-4635-0066

**Abstract:** With the development of big data, analyzing the environmental benefits of transportation systems by artificial intelligence has become a hot issue in recent years. The ground traffic changes can be overlooked from a high-altitude perspective, using the technology of multi-temporal remote sensing change detection. We proposed a novel unsupervised algorithm by combining the image transformation and deep learning method. The new algorithm for remote sensing images is named multi-attention slow feature analysis (ASFA). In this model, three parts perform different functions respectively. The first part records to the K-BoVW to classify the categories of the ground objects as a channel parameter. The second part is a residual convolution with multiple attention mechanisms including temporal, spatial, and channel attention. Feature extraction and updating are completed at this link. Finally, we put the updated features in the slow feature analysis to highlight the variant components which we want and then generate the change map visually. Experiments on three very high-resolution datasets verified that the ASFA has a better performance than four basic change detection algorithms and an improved SFA algorithm. More importantly, this model works well for traffic road detection and helps us analyze the environmental benefits of traffic changes.

**Keywords:** remote sensing; unsupervised change detection; deep learning; slow feature analysis; multi-attention

## 1. Introduction

The transport system industry plays a critical role in the impact on the environment. The connectivity of transportation not only connects the economy but also affects the environmental benefits [1]. For better sustainable development, the monitoring of the traffic environment is crucial. Artificial intelligence and remote sensing technology provide strong support for this, which can save workforce and material resources in order to achieve automated change detection.

Change detection for remote sensing is a task that can obtain the change information in a certain way by comparing images of the same area at different times. It helps us to analyze the reason or trend for the changes and propose targeted countermeasures. Change detection in the field of transportation under the background of artificial intelligence can help us monitor the transportation system, including road change types, change areas and change details. More importantly, it helps professionals measure the impact of carbon emissions [2].

Compared with the traditional method, the multi-temporal change detection [3] focuses more on the analysis of the change law on the time scale. It is a change in a long-term

sequence. With the improvement of the spatial, temporal, and spectral resolution of remote sensing images, the changes in images will contain more in-depth information, guiding the direction of human production and life. Changed information can help us in many fields such as the environmental benefit analysis, restoration of arable land and grassland, the replacement of forest growth, the reasons for natural disasters, etc.

The methods of change detection can be divided into six parts, including direct comparison, image transformation, machine learning, object-oriented, deep learning, and other special algorithms. The direct comparison method usually finds the band characteristics corresponding to the remote sensing images through pixel-level comparison [3] and uses the algebraic difference or ratio method to determine the change. Under the guidance of this kind of thinking, scholars have proposed Change Vector Analysis (CVA) [4], which corresponds to the intensity and type of change through the length and direction of the vector. CVA is currently one of the most widely used algorithms in the field of change detection, and it contributes a basic basis for the development of this field [5,6]. Image transformation is based on the direct comparison of pixels, highlighting the changes of features, showing the types of changes, and improving the accuracy and effect of detection through the spatial transformation of the image. The earliest used image transformation method is Principal Component Analysis (PCA) [7], which separates the extraction of features and difference calculations, and extracts the data features for change detection. Slow Feature Analysis (SFA) [8] has also been proven to highlight the changing information by finding the amount of slowly changing information in the change, grasping the changing area, and further optimizing the detection accuracy.

The birth of machine learning aroused the upsurge of artificial intelligence, and the change detection algorithm based on artificial neural networks and the support vector machine emerged stage by stage. Experiments have proved that using the support vector machine classifier to analyze the changing area can improve the accuracy of change detection [9,10]. Compared with the previous pixel-based detection methods, an object-oriented change detection processing unit is a specific object, which needs to consider the characteristics of the object information, including spectral features, spatial features, and texture features. Generally, it is necessary to segment multiple objects in the image separately, and then perform change detection on objects in different time phases. Some scholars have proposed object-based detection methods [11,12], which have been well verified. After the advent of deep learning, the change detection method based on deep learning is the hottest research direction. Deep learning relies on deep artificial neural networks, and the process of learning and training requires a lot of data support. The deep learning method is characterized by a high degree of automation [13,14].

## 2. Related Works

Slow feature analysis [8] is an unsupervised deep learning method proposed by Wiskott in 2002, which extracts slowly changing features by determining linear input features. SFA can find out the slowly changing signals in the rapidly changing time series, and sort the corresponding eigenvalues, to infer the target object with real significant changes in the opposite direction. It is very suitable for the time series, such as the change detection of multi-temporal and long-term sequences, and has been widely used in human action recognition and dynamic images or video recognition. Therefore, we can use the SFA algorithm to extract the eigenvalues of the transformed linear characteristics, find the signals with slow changes, and reversely highlight the changing scene space, to determine the change information we want. Accordingly, the slow feature analysis applies to the case with slow changes in the background. At the same time, the slow feature analysis algorithm does not need label data and can realize unsupervised feature learning, which is more suitable for the detection with fewer label data.

In the remote sensing change detection images, the pixels in the invariant region correspond to the slow change feature extracted by SFA. On the other hand, the changed pixels correspond to the feature that changed rapidly. Accordingly, we can reduce the

attention to the invariant region by weakening the characteristic value of slow change in the slow feature analysis, and reversely highlight the attention to the changing region to obtain the difference map. However, the feature extraction in the slow feature analysis algorithm has limitations such as linear analysis, which will be deficient in solving the problems in nonlinear space. Therefore, many scholars had improved the algorithm. In 2014, Wu et al. proposed three improved slow feature analysis algorithms [15]. USFA [15] was an unsupervised slow feature analysis method that took the whole remote sensing images as the input of the model to obtain the projection matrix. If the real change seemed slight, the detection effect would be a little poor. This method was suitable for applications with very obvious changes, such as anomaly detection. SSFA [15] was a supervised slow feature analysis method. Unchanged pixels were manually selected through label information for learning and training, which was suitable for ground object detection with label information, but a lot of manual operations were still needed. ISFA [15] was a slow feature analysis method for iterative weighting, which improved the weight of unchanged pixels, reduced the weight of changed pixels, and then trained the projection matrix. In 2017, the team of *Chen Wu* also proposed a kernel-based slow feature analysis algorithm KSFA [16], which used the bag of words model and Bayesian classifier in scene change detection. It has good results in bi-temporal scene detection. At the same time, it also verified that the slow feature analysis has nonlinear expansion ability, which could improve the complexity of feature representation in this regard. However, the performance depended on the selection of kernel function. DSFA [17] was a slow feature analysis algorithm based on deep networks proposed by Du et al. in 2019. Using the idea of suppressing invariant components and highlighting changing pixels, the CVA was used to pre-classify the detection samples, and then the original input features were projected to the high-dimensional feature space through the deep network so that the improved SFA was no longer limited by linearity. Many experiments proved that DSFA has a good effect on detection. In 2021, He et al. proposed a single-band slow feature analysis method [18]. Each band was extracted for slow feature analysis, and the change region was obtained by a filter and clustering method.

The visual bag of words model (BoVW) [19,20] was originally applied to the retrieval of text information. All the words in the text were put in a bag. Therefore, the model was called bag of words. It did not need to consider the order and grammatical structure of words in the text but only cared about the frequency of words. In multiple texts, the dictionary could be constructed according to the number of words appearing in all texts. The number of non-repetitive words was the dimension of the dictionary; in addition, the serial number was the index. The number of duplicates of each independent word was the corresponding value of the index. In the application of images, the visual bag of words could well represent image features for classification by extracting features to calculate probability and obtain more detailed image features. Therefore, taking BoVW as the pre-classification of unsupervised slow feature change detection could improve the detection effect of SFA to a certain extent.

The rapid development of big data and deep learning makes the neural network has a strong ability including training of models and learning of features [21,22]. However, with the increase in the complexity of the model and the amount of information, the optimization of the algorithm is still a key problem to be solved. The attention mechanism drew lessons of human brain information processing which pays attention to local key details. This way helped the neural network model to improve memory ability [23,24]. It could filter out the collection of accurate and effective information from massive redundant information, and also improve the performance of information processing. The introduction of the attention mechanism in change detection could focus on local effective features in a wide range of images and the different changes in images. This method could obtain real and practical information in images and effectively improve the performance of the network model.

Given the above ideas, we propose a multi-attention slow feature analysis (ASFA) model used in transportation. The main contributions of this study could be summarized as follows:

(1) A new unsupervised high-resolution multi-temporal remote sensing change detection algorithm for traffic changes called ASFA is proposed. This model introduces the idea of classification before training and generates the number of categories in images. The residual neural network with multiple attention mechanisms is used to extract complex features, which will be the input data of the slow feature analysis network. The change results will be generated by the threshold algorithm at last.

(2) Four experiment strategies are proposed including the STANet model [21], bi-attention SFA, channel attention SFA, and ASFA. A large number of comparative experiments on three standards of very high resolution (VHR) datasets are used to verify the accuracy and performance of the ASFA model, which is more suitable for the current situation.

(3) A very high-resolution test dataset ZMP-CD is proposed to test the performance and ability of application migration, which proves the value of the model in practical application.

(4) The environmental benefits are analyzed and predicted through experimental data from the model we proposed, which can provide a scientific basis for sustainable development and energy application.

## 3. Materials and Methods

In this study, an unsupervised multi-temporal remote sensing change detection model based on slow feature analysis is proposed. Firstly, the K-BoVW model is used to pre-classify the multi-temporal remote sensing images to obtain the types of ground objects. The following parameter in channel attention is obtained in this part. Secondly, the attention mechanism with channel and temporal-spatial aimed to calculate the weights of multi-attention. The image features are learned through the residual network and updated by the weights. Finally, the slow feature analysis of multi-attention is used to filter the changes rapidly and generate the change detection results by the threshold algorithm.

### 3.1. Slow Feature Analysis

In remote sensing change detection, slow feature analysis is verified to be a better method based on image transformation. The slow feature analysis assumes that the main sensing signals from local attribute coding change rapidly, while the environment changes change slowly [8]. The goal to be studied is not strictly invariant ones but the pixels that change slowly. The definition of SFA can be expressed by the optimization problem:

$$x(t) = [x_1(t), x_2(t), x_3(t), \cdots, x_m(t)]^T \tag{1}$$

$$f(x) = [f_1(x), f_2(x), f_3(x), \cdots, f_n(x)]^T \tag{2}$$

where $x(t)$ is the given multi-dimensional time correlation input signal, $m$ represents the dimension, and the range of time $t$ is between $t1$ and $t2$, along with $n$ representing the number of slow feature analysis functions.

We want to find a nonlinear function $f(x)$ so that the changes of each component of $y(t) = f(x(t))$ are as slow as possible, which means the change rate is the smallest. The square mean of the first-order derivate of time can be used as the measurement standard for the speed of change [8,17]. This problem can be transformed into a problem of solving the output variation difference who has the minimum value. The first-order derivate of the nonlinear function which respects time $t$ can be solved and the mean value of its square can be obtained to minimize the time variance of the output signal:

$$min\Delta(y_i) : \langle (\dot{y_i}(t))^2 \rangle, i \in [1, 2, 3, \cdots, n] \tag{3}$$

Under the following constraints:

$$\langle y_i(t) \rangle = 0 \tag{4}$$

$$\langle (y_i(t))^2 \rangle = 1 \tag{5}$$

$$\langle y_i(t) y_j(t) \rangle = 0, \forall j < i \tag{6}$$

where $\dot{y}_i(t)$ represents the first-order derivate of $y_i(t)$, and it describes the mean value in time. Formula (3) means our optimization objective and Formula (4) defines the zero mean constraints to simplify the following ones. Formula (5) provides a constraint that the transformed output signal should contain some change information. Formula (6) provides a constraint that the output components are not correlated and contain information of different feature types. It shows the slowest change features that are not correlated with other components, respectively. At the same time, it can sort the similar feature components of the output as well as the first output changes slowest. Under general linear conditions, the transformation function can be expressed as:

$$\langle f_i(x) \rangle = w_i^T x \tag{7}$$

where $x$ is the input signal and $w$ is the transformation vector. $w_i^T x$ is the transpose of $w_i$, so the objective function and constraint conditions can be reorganized as Formula (8):

$$\langle ([w]_i^T x)^2 \rangle_t = w_i^T \langle [\dot{x}\dot{x}]^T \rangle_t w_i = w_i^T A w_i \tag{8}$$

$$\langle (w_i^T x) \rangle_t = 0 \tag{9}$$

$$\langle (w_i^T x)(w_i^T x) \rangle_t = w_i^T \langle [xx]^T \rangle_t w_i = w_i^T B w_i = 1 \tag{10}$$

$$\langle (w_i^T x)(w_i^T x) \rangle_t = w_j^T \langle [xx]^T \rangle_t w_i = w_j^T B w_i = 0 \tag{11}$$

In (8), $A = \langle \dot{x}\dot{x}^T \rangle_t$ represents the expectation of the first-order derivate covariance matrix of the input signal. In (10), $B = \langle xx^T \rangle_t$ is the expectation of covariance matrix of the original input signal. Formulas (9)–(11) corresponds to the three constraints in the original Formulas (4)–(6), respectively. The output signal is sorted according to the corresponding eigenvalues from small to large, and the slowest change area has the smallest eigenvalues. Mathematically, we bring Formula (11) into (8) and obtain the falling formula and the optimization problems can be transformed into generalized eigenvalue problems as follows:

$$\langle (w_i^T x)^2 \rangle_t = w_i^T A w_i = \frac{w_i^T A w_i}{w_i^T B w_i} = \frac{\langle (w_i^T \dot{x})^2 \rangle_t}{\langle (w_i^T x)(w_i^T x) \rangle_t} \tag{12}$$

$$AW = BW\Lambda \tag{13}$$

where $W$ represents the generalized eigenvector matrix, and $\Lambda$ is the diagonal matrix of eigenvalues. The formula just proves that the eigenvalue of the most invariant component with the slowest output signal transformation in the SFA theory is the smallest. In the actual remote sensing change detection, the input signals are often the original pixels' information of multi-temporal images which are relatively separate. Moreover, due to the external factors such as illumination and radiation of the imaging, the difference between the unchanged pixels in the unchanged regions of the two images is not completely the same. Therefore, it is difficult to find a very clear definition of the boundaries between

change and unchanged. Thus, according to the SFA theory, it is necessary to reduce the difference between the unchanged pixels in the transformed feature space to the minimum and suppress the unchanged pixels, so that we can highlight the changed pixel part to make the boundaries obviously. Then, it is easier to use the threshold method to distinguish and detect. Thus, we need to reconstruct change detection model using SFA training like Formulas (14) and (15):

$$min_{w_i} : \frac{1}{n} \sum_{j=1}^{n} \left( w_i^T g_i - w_i^T k_i \right)^2 \tag{14}$$

$$g_i, k_i \in \mathbb{R}^m \tag{15}$$

where $n$ is the total pixel number, $g_i$ represents the pixels of the first temporal, $k_i$ represents the pixels of the second temporal, and $m$ is the number of bands. The constraint condition is rewritten and substituted into the generalized eigenvalue problem as Formulas (16)–(20):

$$\frac{1}{2n} \left[ \sum_{j=1}^{n} w_i^T g_i + \sum_{j=1}^{n} w_i^T k_i \right] = 0 \tag{16}$$

$$\frac{1}{2n} \left[ \sum_{j=1}^{n} (w_i^T g_i)^2 + \sum_{j=1}^{n} (w_i^T k_i)^2 \right] = 1 \tag{17}$$

$$\frac{1}{2n} \left[ \sum_{j=1}^{n} (w_i^T g_i)(w_l^T g_i) + \sum_{j=1}^{n} (w_i^T k_i)(w_l^T k_i) \right] = 0 \tag{18}$$

$$A = \frac{1}{n} \sum_{j=1}^{n} (g_i - k_i)(g_i - k_i)^T \tag{19}$$

$$B = \frac{1}{2n} \sum_{j=1}^{n} g_i g_i^T + \sum_{j=1}^{n} k_i k_i^T \tag{20}$$

When the values of $A$ and $B$ were obtained, the eigenvector matrix $W$ can also be solved. After normalization, the final mapping matrix will be obtained as Formula (21):

$$\hat{w}_i = \frac{w_i}{\sqrt{w_i^T B w_i}} \tag{21}$$

The difference of bi-temporal images mapping matrix is the result of change detection as Formula (22). In addition, the schematic of remote sensing change detection with slow feature analysis is like Figure 1:

$$D_i = \hat{w}^T g_i - \hat{w}^T k_i \tag{22}$$

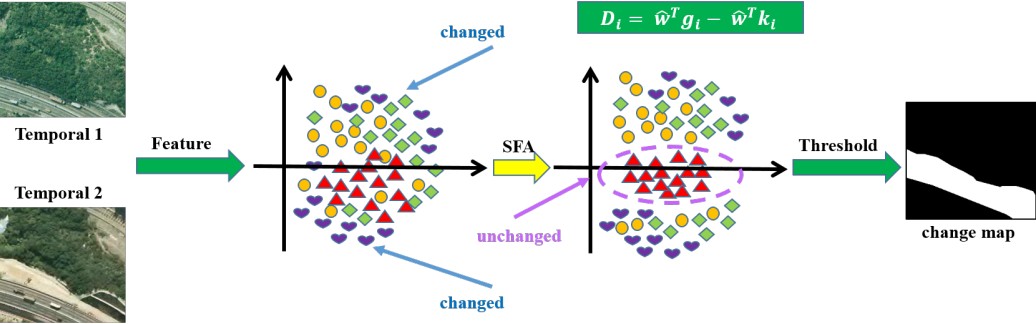

**Figure 1.** The schematic of slow feature analysis for remote sensing change detection.

### 3.2. Multi-Attention SFA

In this section, the unsupervised multi-temporal remote sensing image change detection model ASFA is introduced in detail. Firstly, the multi-temporal remote sensing images are pre-classified by K-BoVW. The visual bag of words model is used to extract the features, and the clustering method is used to classify the number of ground objects in the image (see Section 3.2.1 for details). Secondly, the residual convolution neural network is used to train and generate the feature map. The size and channels of the image will be reset according to the number of categories after clustering through deconvolution and bilinear interpolation, while the weights of channel attention, spatial attention, and temporal attention of multi-temporal images will be calculated, respectively (see Section 3.2.2 for details). Finally, the extracted complex features could be the input values of slow feature analysis. According to the principle of SFA, the invariant components are suppressed to minimize the eigenvalue of the invariant pixels, and then the change region is highlighted. Then, the final change map is obtained by threshold segmentation (see Section 3.2.3 for details).

The input of this model is the processed high-resolution remote sensing images. Firstly, the extended category channel information is obtained through the pre-classification module, and then the original color channels and the category channel are filtered through the convolution network to generate the feature maps. After feature extraction by the second module, the obtained high-dimensional features are transmitted to the slow feature analysis module. The overall structure of ASFA is shown in Figure 2. The specific algorithm of ASFA is shown in Algorithm 1.

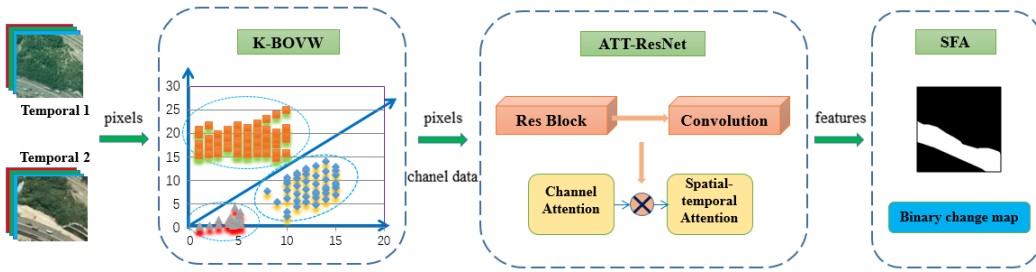

**Figure 2.** The schematic of multi-attention slow feature analysis.

---

**Algorithm 1** Flowchart of ASFA

---

**Input:**
    Multi-temporal input images $X_n^{t1}$, $X_n^{t2}$ ;
    The ground truth $y_n$;
**Output:**
    The binary change map $D$;
  1: Standardize $X_n^{t1}$, $X_n^{t2}$;
  2: Initialize the parameters $\theta_1, \theta_2$;
  3: Employ K-BoVW pre-classification to generate the channel attention mechanism parameter K;
  4: **for** $i < max_e poches$ **do**
  5:     Calculate the features $X_\phi^{t1}$, $X_\phi^{t2}$;
  6:     Extract features $f_n^{t1}$ from $X_n^{t1}$;
  7:     Extract features $f_n^{t2}$ from $X_n^{t2}$;
  8:     Obtain refined feature map $F_{t1}$ and $F_{t2}$;
  9:     Calculate the loss $\mathcal{L}_{ASFA}$ by Formula (23);
 10:     Calculate the gradient and update the parameters with Gradient Descent algorithm;
 11:     $i + +$;
 12: **end for**
 13: Calculate the different maps by Formula (22);
 14: Threshold to obtain the binary change map $D$;
 15: **return** $D$.

---

### 3.2.1. Unsupervised Pre-Classification

Accurate image classification needs complete semantic label information, but annotation of remote sensing images needs professional technical knowledge and a large number of human resources. Thus, it is difficult to obtain remote sensing datasets with semantic labels. Therefore, our model uses an unsupervised pre-classification method to realize the number of categories of ground objects in remote sensing images. We use the visual bag of words model and K-means classifier to complete the task and obtain the training parameter in the next multi-attention residual convolution network.

In the field of image recognition, the BoVW model has obvious advantages in the application of scene classification [20]. However, the images can be seen as a combination of texts, while the image features can be seen as words in the text. BoVW for image classification includes two steps. First, the feature vector of the image is generated by feature extraction; after that, the classifier is used for image classification. The most widely used feature extraction algorithm is Scale-invariant feature transform (SIFT) [25], which was first published by David Lowe of Columbia University in the UK. This feature description method can extract local features with high performance and shield noise, which is very suitable for the classification of scene images.

In traditional change detection, the datasets with semantic labels can help to classify objects by classifier, but in remote sensing, the datasets are difficult to obtain. Therefore, in our model, the k-means clustering method helps us classify the features in an unsupervised way, and obtains the number k of objects without knowing the detail type of the image scenes. We can predict the range of k in the unsupervised datasets to determine the threshold of the k value. Multiple k-means clustering cycles will be done to obtain the value k, which contributes the optimal classify result that we call K-BoVW.

### 3.2.2. ResNet with Attention Mechanism

The model proposed in this paper select the residual network (ResNet) [26] which is suitable for a change detection task depending on the ability to extract complex features [21]. The ResNet was proposed by *He* in 2015, and it broke the depth limit of neural network training aiming at the degradation phenomenon in the network. The attention mechanism aims to pay attention to the importance of a certain local feature in the global range. It is a problem to calculate the probability of attention distribution for key vectors and to calculate the weight value of local feature attention. In addition, the attention mechanism helps to select the important information transmitted to the neural network training to reduce the network burden. The model of CBAM [27] proposed that an attention mechanism based on space and channel can be integrated into any convolutional neural network for training to improve performance [27]. The attention weight could be calculated more effectively by combining the attention mechanism of two dimensions in a series. The feature channel attention mechanism focuses on what is useful information, while the spatial attention mechanism focuses on the position of useful information—which is what we need in remote sensing.

Therefore, we introduce a multi-attention mechanism in the ASFA model and add the attention weight in the three dimensions of channel, spatial and temporal. In the research of spatial-temporal attention module [21] proposed by *Chen*, it proved that the pyramid spatial-temporal attention module has the best performance. The STANet [21] model and CBAM model have verified the superiority of the attention mechanism, but few people combine the training characteristics of the three kinds of attention mechanisms. Therefore, we measured the attention mechanism in three dimensions in the ResNet.

Usually, the channel refers to the color channel of remote sensing images, which represents feature data of input images. In the previous pre-classification, we have obtained the approximate number of categories as a new channel to expand the features. The channel acquisition needs convolution operation, so the channel of this model needs four convolution filters to generate. We could obtain the feature maps and input them into the ResNet. In our experiments, we selected high resolution RGB images in png format.

Because the size and number of channels of the feature map generated by the residual network have changed, we use inverse convolution and interpolation methods to achieve up-sampling [28,29] to improve the resolution of the image. Thus, we could obtain the number of channels and image size we want. This operation could obtain a better training effect; then, the attention model is used to train and calculate the multi-dimensional attention weight to obtain more accurate image features.

In this model, the adaptive channel attention mechanism is to calculate the three channels of RGB in parallel, and the input data are the image of $\mathbb{R}^{C\times H\times W\times K}$. The adaptive multiple attention mechanism is divided into four parts, and we drew a lesson from the PAM in STANet [21]. Each part divides the feature vector into several sub-regions of specific scales on average to obtain the local attention representation, and then the four results are aggregated to obtain the final feature. The structure of the multiple attention is shown in Figure 3, and the structure of the ResNet with Channel attention is shown in Figure 4.

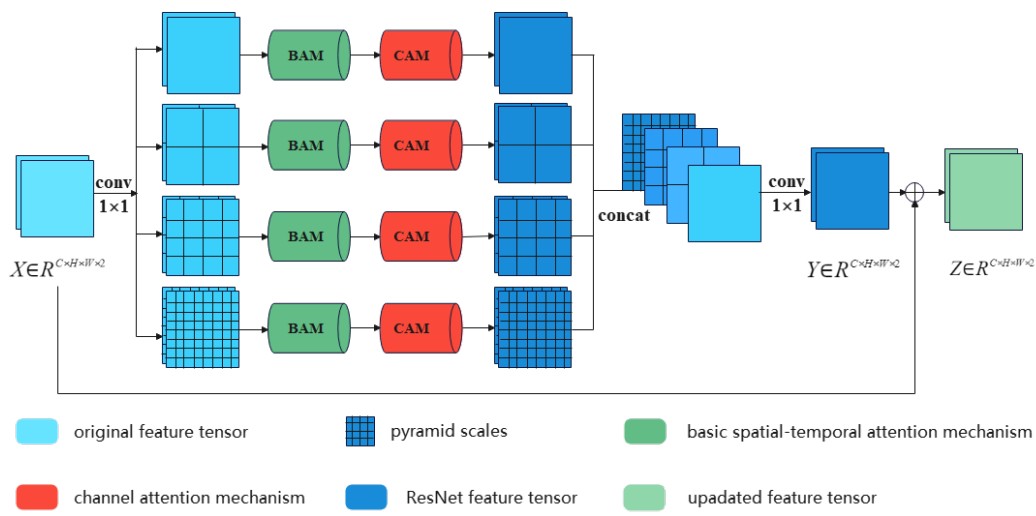

**Figure 3.** The structures of multi-attention.

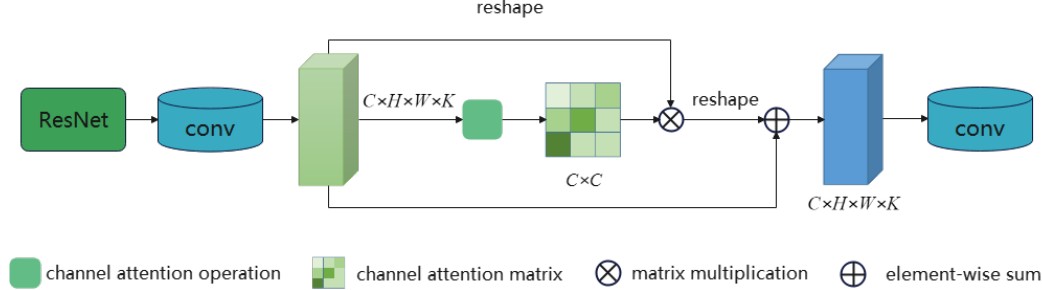

**Figure 4.** The structures of ResNet with Channel attention.

### 3.2.3. Multi-Attention Slow Feature Analysis

The complex image features are introduced into the slow feature analysis model as input. Through the optimization of the objective function, the eigenvalue of the invariant component is minimized, and then the variable and invariant pixels are distinguished. The final change map is obtained by using the threshold algorithm.

According to the SFA principle, the pixel with the slowest change has the smallest eigenvalue, which inhibits the pixels with the slowest change and makes the change pixels easier to distinguish. Therefore, our goal is to minimize the eigenvalue of the pixel with the slowest change.

The loss function of ASFA we set is shown in Formula (23) based on the theories of DSFA [17] and STANet [21]. Formulas (24)–(27) are decomposition of Formula (23):

$$\mathcal{L}_{ASFA}(D^*, M^*, \theta_1, \theta_2) = \mathcal{L}_1 + \mathcal{L}_2 \tag{23}$$

$$\mathcal{L}_1 = \frac{1}{2}\frac{1}{n_u}\sum_{b,i,j}(1 - M^*_{b,i,j})D^*_{b,i,j}$$
$$+\frac{1}{2}\frac{1}{n_c}\sum_{b,i,j}M^*_{b,i,j}Max(0, m - D^*_{b,i,j}) \tag{24}$$

$$\mathcal{L}_2 = tr\left[(B_\phi^{-1}A_\phi)^2\right] \tag{25}$$

$$n_u = \sum_{b,i,j}(1 - M^*_{b,i,j}) \tag{26}$$

$$n_c = \sum_{b,i,j}M^*_{b,i,j} \tag{27}$$

In (23), $tr(\cdot)$ means the trace of a matrix, $\theta_1, \theta_2$ are the parameters obtained by the gradient optimization [17]. $D^*$ and $M^*$ are the batch of distance maps and binary label maps, respectively, where 0 represents no change and 1 denotes a change. $b$, $i$, and $j$ denote the indexes of the batch, height, and width, respectively. $n_u$ is the number of the no-change pixel pairs; in addition, the changed ones are represented by $n_c$.

Threshold segmentation is a traditional and effective image segmentation method. The algorithm can divide the pixels of the image into multiple categories, which is suitable for segmentation tasks with large gray level differences between the target and the background environment. The OTSU algorithm [30,31] is an adaptive threshold determination method proposed in 1979, and the objective is to minimize the inter-class variance of the separated object and the background image, while the K-means clustering algorithm [32,33] classifies by calculating the distance between the object and the clustering center. Since the k-means clustering method has been used in pre-classification, OTSU was selected as the threshold algorithm for experiments, in order to avoid affecting judgment in different links.

The OTSU divides the image into two parts which are the background and the target, according to the grayscale characteristics of the image. The larger the inter-class variance between the background and the target, the greater the difference between the two parts of the image. These threshold algorithms can accurately segment the categories with obvious differences and are suitable for the segmentation of slow feature analysis. Therefore, the ASFA model used the threshold segmentation method of OTSU to generate the binary change diagram. The change intensity diagram is described by the chi-square distance [17] which can measure the difference between objects. Formula (28) shows the chi-square distance, where $m$ is the number of feature bands, and $\sigma^2$ means the variance of each band:

$$chi2 = \sum_{i=1}^{m}\frac{(D_\phi^i)^2}{\sigma_i^2} \tag{28}$$

## 4. Experiment

### 4.1. Datasets

We selected three public and real high-resolution datasets for training, including SZTAKI [34], SYSU-CD [35], and LEVIR-CD [21]. The SZTAKI AirChange Benchmark set, proposed by Csaba Benedek and Tamás Szirányi, contains three groups of 13 pairs of optical aerial images with the difference of multi-years. The image resolution is $952 \times 640$, and the spatial resolution is 1.5 m. The change scenes in the images include new facilities, buildings, planting trees, arable land, and so on. SYSU-CD was produced by the team of Sun Yat-sen University. Optical aerial photography captured the changes in Hong Kong

and its surrounding areas from 2007 to 2014. The dataset contains 20,000 pairs of small-size images, which are 256 × 256, and the spatial resolution reaches 0.5 m per pixel. It mainly includes new urban buildings, urban expansion, vegetation changes, road traffic expansion, and maritime construction. LEVIR-CD is a large-scale change detection data set published by Chen et al. in 2020. The images are obtained from Google Earth and collected in many regions of Texas, the United States. There are 637 pairs of VHR images with a spatial resolution of 0.5 m, and the size is 1024 × 1024. The period is also relatively large, including 5–14 years.

The three standard datasets are all-optical RGB images and contain binary change labels for verification. The spatial resolutions are all below 1.5 m. The first two datasets are from real aerial images, and the third dataset is made from Google Earth. The datasets we selected cover multiple urban areas as a whole and have the representative ability.

To prove that our model can be migrated to practical applications, we selected images with practical detection significance from Google Earth to make a small testing dataset ZMP-CD. The area we study is the Zhuhai-Macao port in the Hong Kong-Zhuhai-Macao Bridge. Moreover, we select a long time series of images between 2009–2019 that corresponds to the constructive process of the bridge. Finally, we analyzed the environmental benefits brought by the Hong Kong-Zhuhai-Macao Bridge expansion. The detailed description of datasets we selected is given in Table 1, and Figure 5 shows two pairs of images of each dataset for example.

**Table 1.** The description of the datasets.

| Dataset | Resolution (m) | Size (px) | The Type of Changes |
| --- | --- | --- | --- |
| SZTAKI | 1.5 | 952 × 640 | urban change, construction, planting trees, new farmland and change of construction |
| SYSU-CD | 0.5 | 256 × 256 | suburban dilation, groundwork, change of vegetation, road expansion and sea construction |
| LEVIR-CD | 0.5 | 1024 × 1024 | building-related, building growth, building decline and land change |
| ZMP-CD | 0.5 | 1024 × 1024 | construction of Zhuhai-Macao Port in the Hong Kong-Zhuhai-Macao Bridge |

*4.2. Experiment Setting*

In this section, we propose four strategies of the experiment to obtain the best model. Each strategy is trained by the three datasets mentioned above. We will find the optimal strategy by comparing the experimental data.

The first experimental strategy is the change detection model based on spatial and temporal attention mechanism [21]. The second experimental strategy is the slow feature analysis based on the spatial-temporal attention mechanism. The third experimental strategy is the slow feature analysis based on the channel attention mechanism. In addition, the fourth experimental strategy is the ASFA, which is the slow feature analysis based on a multi-attention mechanism.

For all the experiments, we use the Pytorch framework. In these models, the weight and bias matrices of each layer are initialized randomly and optimized afterward. Five unsupervised change detection algorithms introduced in Section 1 are selected for comparative experiments, including CVA [3], PCAKmeans [36], MAD [37], SFA [8], and DSFA [17]. A detailed description of these methods is given in Table 2.

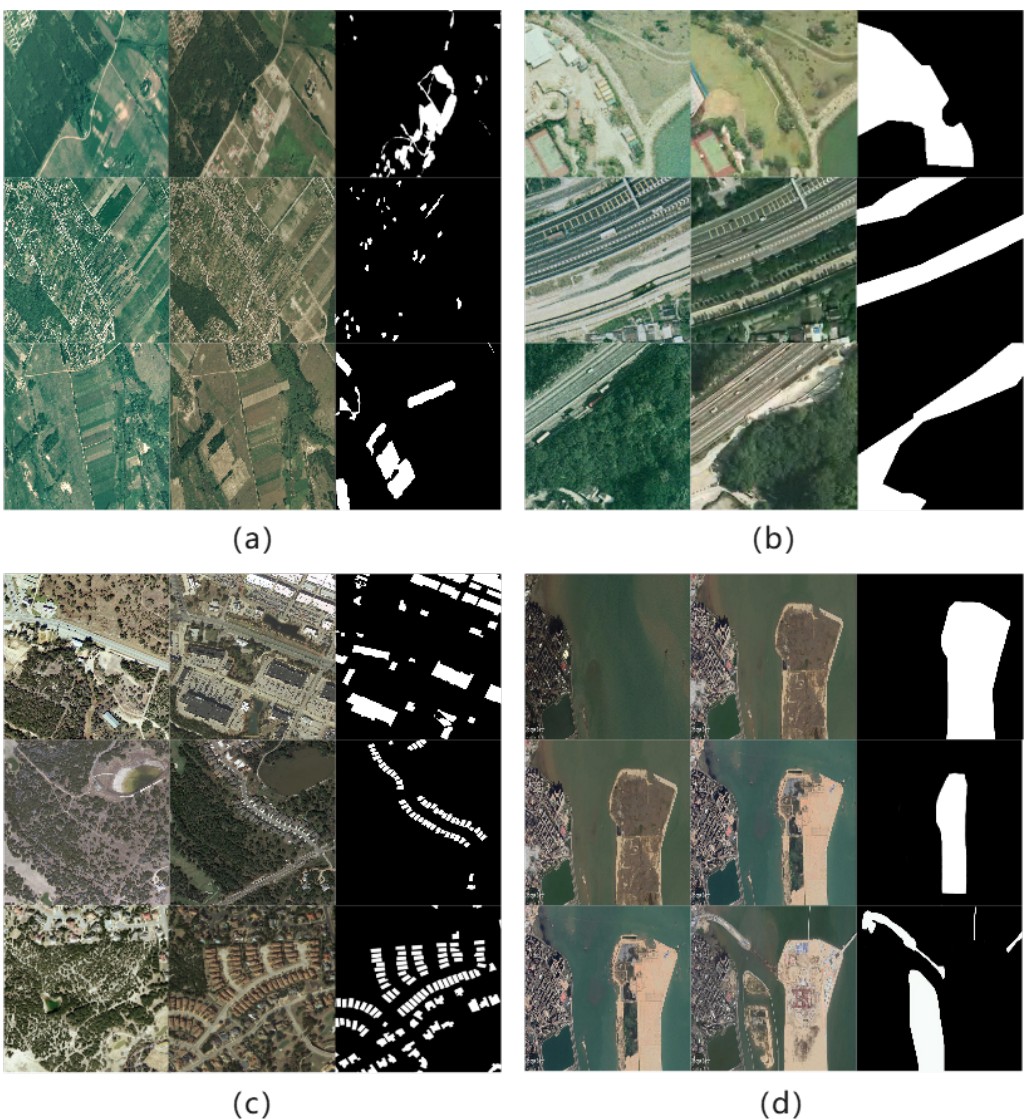

**Figure 5.** The example of four datasets.(**a**) SZTAKI; (**b**) SYSU-CD; (**c**) LEVIR-CD; (**d**) ZMP-CD.

**Table 2.** The description of the other unsupervised change detection method.

| Method | Description |
| --- | --- |
| CVA | use vector difference and change the angle to predict change intensity and category |
| PCAKmeans | Divide the difference image into non-overlapping blocks to extract features by PCA and to create an eigenvector space. |
| MAD | maximize the correlation between the features of multi-temporal images |
| SFA | aim to find the most invariant component in remote sensing images to highlight changed regions |
| DSFA | represent and transform image features by deep networks |

To verify the performance of the model, we choose the following representative metrics, including OA, Mean_IoU, precision, recall, and F1 score. However, most of them are calculated by the basic indicators of positives and negatives, where TP and FP refer to true positives and false positives, and TN and FN refer to true negatives and false negatives. The information of the evaluating metrics is shown in Table 3.

**Table 3.** The evaluating metrics.

| Metrics | Representation and Formulation |
| :---: | :--- |
| OA | the overall accuracy<br>OA = (TP + TN)/(TP + TN + FP + FN) |
| MIoU | the mean intersection over union<br>MIoU = $1/(k+1)\sum_{i=0}^{k}[\text{TP}/(\text{FN}+\text{FP}+\text{TP})]$ |
| precision | the false alarm rate<br>precision = TP/(TP + FP) |
| recall | the miss alarm rate<br>recall = TP/(TP + FN) |
| F1 | consider both precision and recall<br>F1 = $2\times$ precision $\times$ recall/(precision + recall) |

### 4.3. Experiments of Four Strategies

#### 4.3.1. The STANet Model

We use the state-of-the-art method STANet [21] to obtain the detection accuracy of supervised learning using the datasets we selected. This can be a reference to the results of our unsupervised method. The STANet model is famous for spatial-temporal attention. The spatial attention focuses on the information of the location of pixels; in addition, the temporal attention reflects the multi-temporal images. We first extract the feature of the input images using the ResNet. After that, the bi-attention mechanism is used to update the features with the local detail information. Finally, we compare the pixels pairs of the bi-temporal features and generate the distance map. The experiments of the three datasets are shown in Table 4.

**Table 4.** The experiment results of STANet on three standard datasets.

| Dataset | OA | MIoU | Precision | Recall | F1 |
| :---: | :---: | :---: | :---: | :---: | :---: |
| SZTAKI | 0.93659 | 0.68288 | 0.81932 | 0.80395 | 0.81156 |
| SYSU-CD | 0.88262 | 0.73299 | 0.80555 | 0.89055 | 0.84592 |
| LEVIR-CD | 0.92343 | 0.75390 | 0.82101 | 0.90220 | 0.85969 |

#### 4.3.2. SFA Based on Bi-Attention

In this experiment, we use the slow feature analysis with a bi-attention mechanism. We extract the preliminary features by the residual convolution network. In addition, then the features will be updated by the attention with the PAM model [21], which added the local spatial and temporal information. This attention mechanism is suitable for remote sensing images. After obtaining the high-level features, the slow feature analysis model will be worked to separate the changed pixels and the unchanged ones. The features changed slowly represent the unchanged pixels. Finally, the threshold method is used to generate the change map. Table 5 shows the results of SFA based on bi-attention.

**Table 5.** The experiment results of bi-attention SFA on three standard datasets.

| Dataset | OA | MIoU | Precision | Recall | F1 |
| :---: | :---: | :---: | :---: | :---: | :---: |
| SZTAKI | 0.93727 | 0.68549 | 0.82039 | 0.80653 | 0.81340 |
| SYSU-CD | 0.88501 | 0.73758 | 0.80802 | 0.89430 | 0.84897 |
| LEVIR-CD | 0.92286 | 0.75365 | 0.82601 | 0.89588 | 0.85952 |

#### 4.3.3. SFA Based on Channel Attention

In this experiment, we use the slow feature analysis based on the channel attention mechanism. Firstly, we detect the images with the K-BoVW model to obtain the number of object categories. This number will transport into the channel attention mechanism as

a parameter. After that, we extract the preliminary features and update them by channel attention to extend the channels. This attention mechanism focuses on the local channel information. In the end, we complete the change detection with the slow feature analysis model. Table 6 shows the results of SFA based on channel attention.

**Table 6.** The experiment results of channel attention SFA on three standard datasets.

| Dataset | OA | MIoU | Precision | Recall | F1 |
|---------|-----|------|-----------|--------|-----|
| SZTAKI | 0.94023 | 0.69733 | 0.82621 | 0.81721 | 0.82168 |
| SYSU-CD | 0.88920 | 0.74447 | 0.80701 | 0.90572 | 0.85352 |
| LEVIR-CD | 0.92271 | 0.75172 | 0.81902 | 0.90149 | 0.85827 |

### 4.3.4. SFA Based on Multi-Attention

In the last experiment, we use the ASFA, which is the slow feature analysis based on the multi-attention mechanism. We use the K-BoVW model to obtain the parameter of the channel attention mechanism. After obtaining the features from the network, we update them with multi-attention including temporal, spatial, and channel. Finally, we put the last features to the slow feature analysis model to obtain the change result. Table 7 shows the results of SFA based on multi-attention.

**Table 7.** The experiment results of multi-attention SFA on three standard datasets.

| Dataset | OA | MIoU | Precision | Recall | F1 |
|---------|-----|------|-----------|--------|-----|
| SZTAKI | 0.95310 | 0.74639 | 0.82826 | 0.88306 | 0.85478 |
| SYSU-CD | 0.89280 | 0.75174 | 0.81150 | 0.91077 | 0.85828 |
| LEVIR-CD | 0.92357 | 0.75448 | 0.82201 | 0.90181 | 0.86006 |

According to the three data sets to evaluate the indicators, the experimental data can be summarized in Tables 8–10. The accuracy of the ASFA can be compared with the supervised algorithm and even better. The best overall accuracy reaches 95 percent.

**Table 8.** The experiment results on the SZTAKI dataset with four strategies.

| Strategy | OA | MIoU | Precision | Recall | F1 |
|----------|-----|------|-----------|--------|-----|
| STANet | 0.93659 | 0.68288 | 0.81932 | 0.80395 | 0.81156 |
| bi-attention SFA | 0.93727 | 0.68549 | 0.82039 | 0.80653 | 0.81340 |
| channel attention SFA | 0.94023 | 0.69733 | 0.82620 | 0.81721 | 0.82168 |
| multi-attention SFA | 0.95310 | 0.74640 | 0.82826 | 0.88306 | 0.85478 |

**Table 9.** The experiment results on the SYSU-CD dataset with four strategies.

| Strategy | OA | MIoU | Precision | Recall | F1 |
|----------|-----|------|-----------|--------|-----|
| STANet | 0.88262 | 0.73299 | 0.80555 | 0.89055 | 0.84592 |
| bi-attention SFA | 0.88501 | 0.73758 | 0.80802 | 0.89430 | 0.84897 |
| channel attention SFA | 0.88920 | 0.74447 | 0.80701 | 0.90572 | 0.85352 |
| multi-attention SFA | 0.89280 | 0.75174 | 0.81150 | 0.91077 | 0.85828 |

**Table 10.** The experiment results on the LEVIR-CD dataset with four strategies.

| Strategy | OA | MIoU | Precision | Recall | F1 |
|----------|-----|------|-----------|--------|-----|
| STANet | 0.92343 | 0.75390 | 0.82101 | 0.90220 | 0.85969 |
| bi-attention SFA | 0.92286 | 0.75365 | 0.82601 | 0.89589 | 0.85952 |
| channel attention SFA | 0.92274 | 0.75172 | 0.81901 | 0.90149 | 0.85827 |
| multi-attention SFA | 0.92357 | 0.75448 | 0.82201 | 0.90181 | 0.86006 |

### 4.4. Experiments on Three Datasets

Through the result of the above experiments, we proved that ASFA has the best performance on three standard datasets. In this section, we compared ASFA with the traditional and deep learning algorithms to verify the reliability of the model. The comparison algorithms we choose are all the unsupervised change detection methods we introduced in Section 3.2. We still choose the three standard datasets as our experimental data. The experiment results and change maps are shown as follows.

Table 11 shows the metrics results of the SZTAKI, and Figure 6 shows the parts' results of the experiments. There is very little change in the SZTAKI air change benchmark, and we detected more change information than the labeled ground truth. It can be seen from the last two sets of pictures that there is almost no change in them. However, we still detected the other changes because color changes can easily be thought of as changes in features. In the experiment, the traditional algorithms are susceptible to seasonal changes and detected some unchanged areas as the target. While the deep learning method is more accurate to avoid the influence of the environment, the ASFA algorithm detected more true samples than the DSFA.

**Table 11.** The comparative experiment results on SZTAKI.

| Method | OA | MIoU | Precision | Recall | F1 |
|---|---|---|---|---|---|
| CVA | 0.93277 | 0.65842 | 0.77759 | 0.81118 | 0.79403 |
| PCAKmeans | 0.93556 | 0.67150 | 0.79036 | 0.81703 | 0.80347 |
| MAD | 0.93266 | 0.65877 | 0.77998 | 0.80913 | 0.79429 |
| SFA | 0.94290 | 0.70024 | 0.80037 | 0.84842 | 0.82369 |
| DSFA | 0.94604 | 0.71644 | 0.81803 | 0.85227 | 0.83480 |
| ASFA | 0.95310 | 0.74639 | 0.82826 | 0.88306 | 0.85478 |

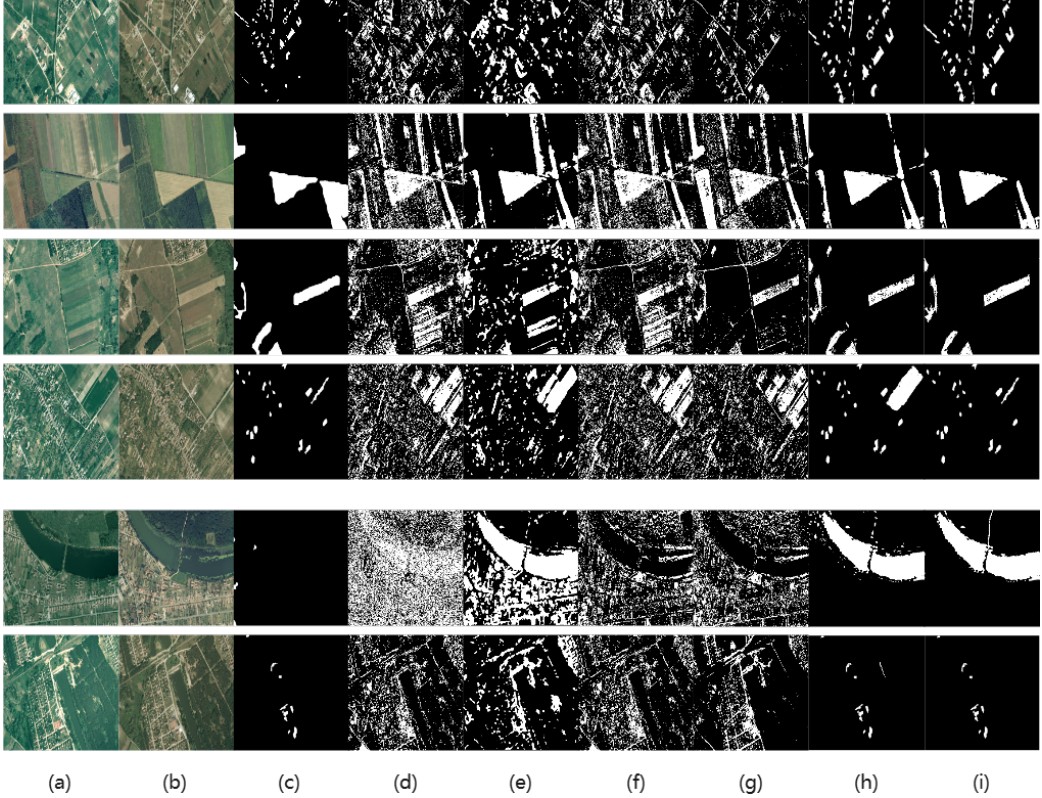

**Figure 6.** The change maps on SZTAKI. (**a**) image 1; (**b**) image 2; (**c**) ground truth; (**d**) CVA; (**e**) PCAKmeans; (**f**) MAD; (**g**) SFA; (**h**) DSFA; (**i**) ASFA.

Table 12 shows the metrics results of the SYSU-CD and Figure 7 shows the parts' results of the experiments. In the SYSU-CD, the labels simply mark the major changes and ignore minor changes such as the tree branches and leaves. We can find the change with our eyes in the last two sets of pictures, which are ignored by the ground truth. Our experiments detected the detail changes, even the change of trees. It can be seen from the data that the algorithm detection improved by deep learning is more accurate. From the image, the boundary detected by our proposed algorithm is more rounded.

**Table 12.** The comparative experiment results on SYSU-CD.

| Method | OA | MIoU | Precision | Recall | F1 |
|---|---|---|---|---|---|
| CVA | 0.86580 | 0.69458 | 0.76301 | 0.88567 | 0.81977 |
| PCAKmeans | 0.88280 | 0.73107 | 0.79650 | 0.89898 | 0.84465 |
| MAD | 0.87206 | 0.70733 | 0.77301 | 0.89276 | 0.82858 |
| SFA | 0.88318 | 0.75256 | 0.79995 | 0.89686 | 0.84564 |
| DSFA | 0.88840 | 0.74298 | 0.80650 | 0.90415 | 0.85254 |
| ASFA | 0.89280 | 0.75174 | 0.81150 | 0.91077 | 0.85828 |

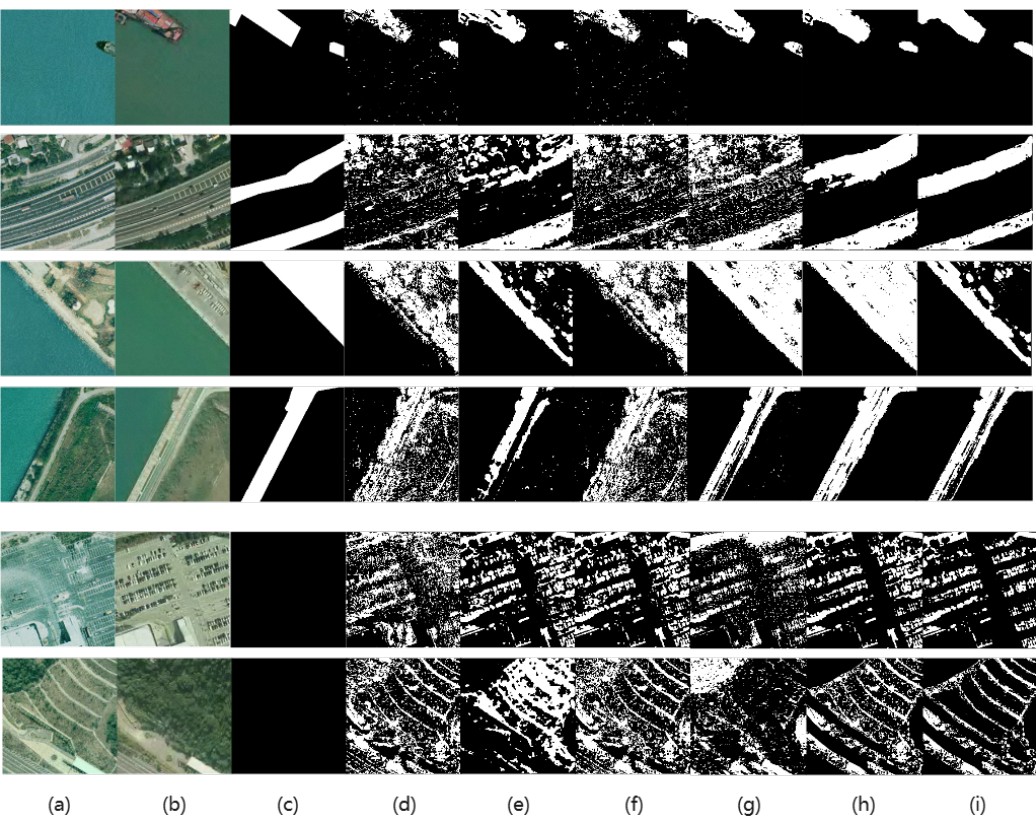

**Figure 7.** The change maps on SYSU-CD. (**a**) image 1; (**b**) image 2; (**c**) ground truth; (**d**) CVA; (**e**) PCAKmeans; (**f**) MAD; (**g**) SFA; (**h**) DSFA; (**i**) ASFA.

Table 13 shows the metrics results of the LEVIR-CD. We can find that the deep learning method was more precise in total. Due to the label of this dataset being simple on the change of the buildings and garages, the value of the ground truth is not exact. Thus, the value of the metrics should be compared horizontally to be more persuasive. From the data point of view, a deep learning algorithm has more advantages than the traditional method. The best indicator data have been highlighted.

**Table 13.** The comparative experiment results on LEVIR-CD.

| Method | OA | MIoU | Precision | Recall | F1 |
|---|---|---|---|---|---|
| CVA | 0.89371 | 0.67340 | 0.76701 | 0.84658 | 0.80483 |
| PCAKmeans | 0.90471 | 0.70276 | 0.78850 | 0.86601 | 0.82544 |
| MAD | 0.90971 | 0.71506 | 0.79301 | 0.87916 | 0.83386 |
| SFA | 0.91371 | 0.72645 | 0.80201 | 0.88521 | 0.84155 |
| DSFA | 0.91986 | 0.74313 | 0.81150 | 0.89817 | 0.85264 |
| ASFA | 0.92357 | 0.75447 | 0.82201 | 0.90181 | 0.86006 |

Figure 8 shows parts' results of the experiments on the LEVIR-CD. We can find that the traditional algorithm did not work very well on the LEVIR-CD dataset with lots of noise through the result maps. Due to the LEVIR-CD aim to determine the change of buildings, the label of this dataset does not focus on the other types of change. Lots of other change information was detected, which was not labeled in the ground truth. As could be seen from the last two sets of pictures, all of the models could detect all the change information, and the detection method of deep learning would be more accurate. All of the models could find out the right changed area and the boundaries of traditional algorithms are blurred; the detection method of deep learning would be more accurate. However, the slow feature analysis has good performance, and the accuracy is also guaranteed compared to other traditional ones. In this case, the methods of deep learning based on SFA were verified to be a correct improvement plan. Compared with the DSFA model, our model has the advantage of obtaining a more precise result in this case.

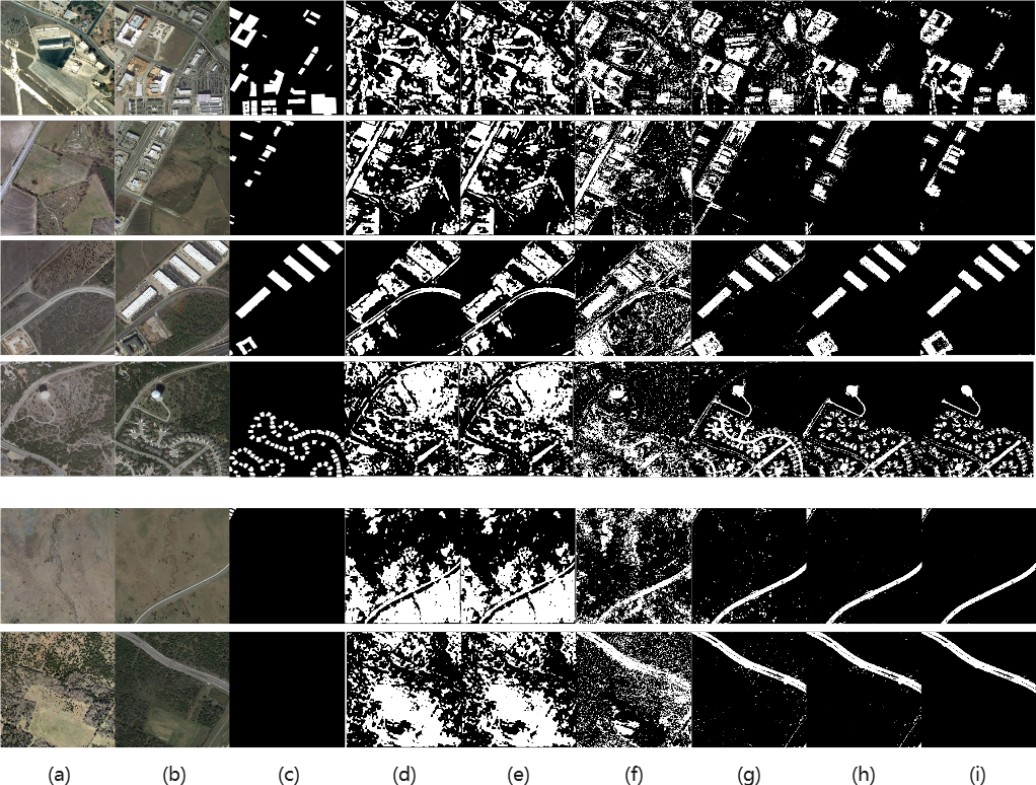

**Figure 8.** The change maps on LEVIR-CD.(**a**) image 1; (**b**) image 2; (**c**) ground truth; (**d**) CVA; (**e**) PCAKmeans; (**f**) MAD; (**g**) SFA; (**h**) DSFA; (**i**) ASFA.

Compared with other unsupervised methods, our algorithm has the advantage in change detection for very high-resolution images.

## 5. Transport in the Carbon Environment

According to the comparison of the above twelve groups of experiments, the overall performance of the ASFA is verified to be the best one. On that occasion, we use the ASFA model to complete the actual change detection task in our ZMP-CD dataset.

We selected the Zhuhai-Macao port of Hong Kong-Zhuhai-Macao Bridge as the study area, which opened to traffic three years ago. The Zhuhai-Macao port is the only domestic port for artificial reclamation at sea with a total area of 2.09 square kilometers. There are so many traffic roads built on the sea which achieve three-way connectivity on this island. Therefore, we can detect more data to analyze the environment of traffic expansion. Due to the traffic of the bridge, the low cost of travel time between the three places is more conducive for business, tourism and other activities. The bridge promotes their economic and social integration. At the same time, it contributes to the region becoming the most dynamic economic zone.

The images in ZMP-CD are collected from Google Earth with a resolution of 0.5 m, and the ground truth is manually marked. The images are shown in the form of long time series, recording the changes of the Hong Kong-Zhuhai-Macao Bridge in the past 10 years. Six high-resolution remote sensing images have been selected from Google Earth every two years since 2009. We construct five sets of change detection pairs to compare the changes of each two years. Our goal is to detect the real change of the Zhuhai-Macao port through the ASFA model in 10 years. We will estimate the change area ratio and the traffic road ratio in change by pixel unmixing [38,39]. In addition, we can analyze the reason for expansion and predict the economic development trend through the diagram.

Figure 9 shows the long time series change maps on the ZMP-CD ASFA model used. Our model is verified to have good migration and application capabilities again, and the following analysis has more reliability and strong persuasion.

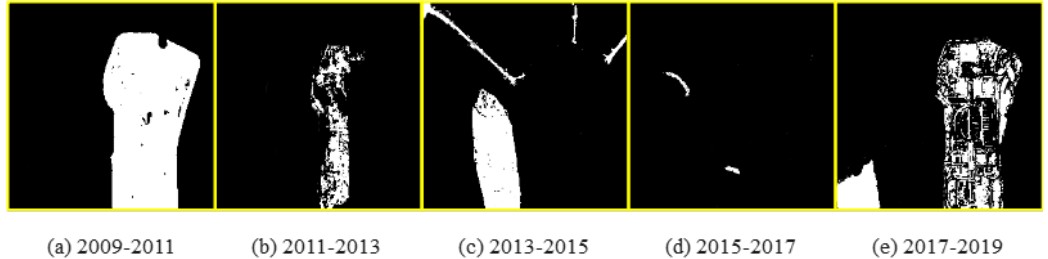

(a) 2009-2011          (b) 2011-2013          (c) 2013-2015          (d) 2015-2017          (e) 2017-2019

**Figure 9.** The long time series change map of ZMP–CD.

Through the analysis of long time series, we can find the change of the bridge construction every two years. We focus on the analysis of changes at Zhuhai-Macao Port. In the first four years, the changing area was almost one-third of the total study area, but without traffic road on the sea. During this period, the artificial island of the Hong Kong-Zhuhai-Macao was mainly constructed. While in the third two years, the ratio of traffic changed area on the sea in the changing area reached 30 percent, and it represents the traffic road expansion beginning during this time. We detected three roads on the sea which were constructed between 2013 and 2015. The Arch North Bay Bridge leads to the Arch North port and the Hong Kong-Zhuhai-Macao Bridge leads to the Hong Kong port [40]. Both of them constitute a part of the Pearl River Delta Ring Expressway. In addition, the other road we detected in the traffic towards the urban of Zhuhai is named Pioneer Road. At the same time, a new district was built near the artificial islands which accounted for more than 70 percent of the change region. Furthermore, the changes in traffic roads we detected took up all the changes in 2017. There were two roads constructed in the new district we talked about above including Dr. Ma Man Kei Avenue and Tun Seng Avenue. These two roads passed through the new district towards Zhuhai-Macao Port and Macao urban, respectively. As well as the last two years, the changes focus on the construction of Zhuhai-Macao Port without the traffic change on the sea. Until 2019, all

expansion and construction were completed. The detailed information of change is shown in Figure 10.

In this experiment to detect selected areas using ASFA, we found five new roads on the sea. We make the area ratio of these roads into a pie chart to help us analyze the proportion of changes. The change ratio of every road we detected is shown in Figure 11. The smallest change is the Tun Seng Avenue connecting the new district and the Zhuhai-Macao port. The proportion of it reached only 2 percent, which was difficult to detect. However, the largest change in our study area referred to the Arch North Bay Bridge, and it occupied the mean seat. All of these proved the detection effect of our model.

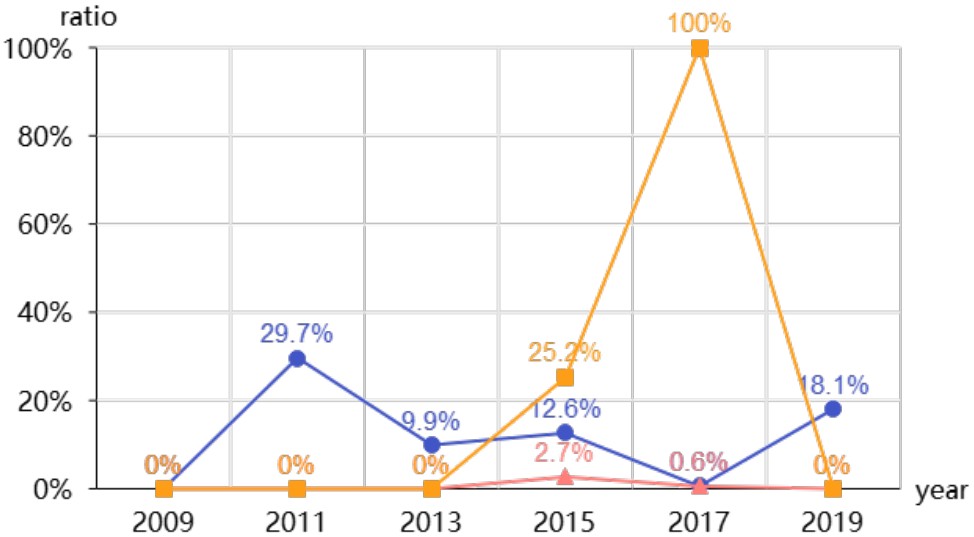

**Figure 10.** The analysis of change detection of the Zhuhai-Macao port.

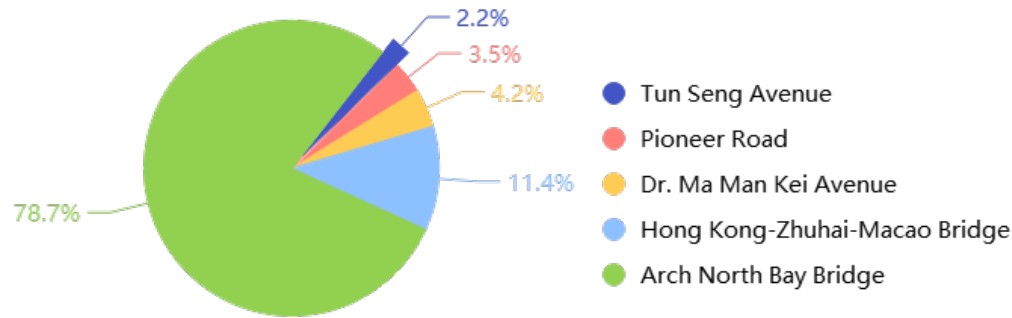

**Figure 11.** The change ratio of traffic expansion on the sea.

We identified the study area as remote sensing mapping by ArcGIS. Figure 12 shows the Hong Kong-Zhuhai-Macao Bridge connected three places and the changed traffic roads in the study area. Five changed roads are represented in red corresponding to the above analysis.

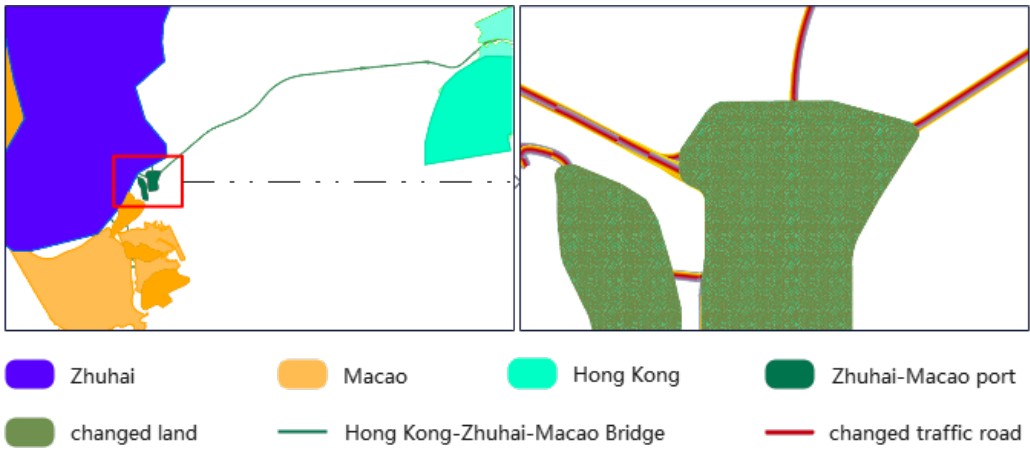

**Figure 12.** The mapping of the study area.

## 6. Results

We set up four experiment strategies to prove the performance of ASFA. Firstly, the supervised detection method STANet with high accuracy is tested as a comparison. The other three are unsupervised improved algorithms based on SFA. The bi-attention model adds a spatial-temporal attention mechanism; in addition, the channel attention model updates the channel parameter in convolution. Finally, the ASFA mixes three attention mechanisms to update the extracted features. After comparison, it is found that the accuracy of unsupervised ASFA can be compared with that of the supervised algorithm, or even better. By comparing ASFA with other traditional algorithms and deep learning ones, it can be seen that the performance of our improved algorithm ASFA is better. Through the test dataset of ZMP-CD, the change information of the Hong Kong-Zhuhai-Macao Bridge in the past 10 years is analyzed. Furthermore, we can analyze the environmental benefits of traffic construction through long-time series images and remote sensing mapping. It further proves that the ASFA model proposed by us has a better ability to migrate other scene detection even the transport system, and it has a reliable ability in the energy application.

## 7. Discussion

The traffic detection system based on big data and artificial intelligence still has many limitations. For example, there is no suitable detection model, the detection lacks relevance, and the amount of data cannot be accurately analyzed. The unsupervised model proposed by us can predict the changes in traffic roads, and analyze the traffic construction and traffic environment through detection data. Through the verification of the ZMP-CD dataset, our model can well detect the impact of road traffic development in the real environment, which is helpful to analyze the balance between economic development and environmental benefits. Combining the change maps and remote sensing mapping, the model has a good application value, which is helpful for the further scientific planning of the traffic management department.

Remote sensing change detection is limited by lacking labeled public datasets. The detection accuracy of high-resolution images still needs to be improved, and the generalization ability of the model is not strong. To break these limitations and obtain better detection models, we propose an unsupervised change detection model for high-resolution images and apply it to the actual tasks with detection and analysis. Through large numbers of experimental data analyses, the superiority of the model is verified, and the significance of this model is explained below.

SFA has a good effect on change detection, and it is good at unsupervised detection. It highlights the change component by suppressing the invariant component, to distinguish the change and unchangeable regions. It can filter out the slowly changing scenes in the image, such as the slow change of the forest over time, effectively filtering out the

unimportant change information and enhancing the truly effective change information. Therefore, many scholars have improved the SFA, but our model is more targeted at high-resolution image detection.

The K-BoVW in this model is indispensable. BoVW is good at the classification of object categories combined with a classifier. It uses a SIFT algorithm to obtain features by calculating frequency. However, the classification is related to the datasets which had been labeled by semantics. Thus, to obtain more meaningful features, we had the idea of obtaining the number k of the categories without detailed labels by the k-means clustering method. We used the RGB images with high resolution to train the model; on the other hand, the images have a shortage of number of channels. Therefore, in the channel attention mechanism, we added parameter k to join the convolution network to have complexity channel parameters. On that occasion, we obtained a better result and verified by the fourth experiment.

Channel attention helps the network to learn more complex features. In addition, the RGB image is short for the number of channels, and the features we extracted are limited. According to the result of K-BoVW above, we obtained the numbers of the categories of the ground objects. Thus, we can expand our channels in the convolution network to learn the high-level features. With the result of the experiments, we find that the addition of channel attention plays a role in the whole model. This algorithm can enhance the ability to determine the detailed information and obtain the local attention representation.

Even though the ASFA model has good behavior, it is still a pity that the adaption of the parameter detection has not been realized yet. Thus, we are going to find a super method to classify the types of objects in the images automatically and prove more adaptive detection models. In addition, we will study scene change detection for remote sensing.

## 8. Conclusions

In this paper, we provided an unsupervised multi-temporal change detection algorithm for the remote sensing transport system to help the traffic monitor. By proposing a multi-attention slow feature analysis based on pre-classification, the detection model is verified to improve in performance. We proposed that the pre-classification method rely on K-BoVW and obtain the parameter k, which is calculated by the k-means clustering algorithm and represents the number of categories in images. We added channel attention with spatial-temporal attention into ResNet to obtain the feature precisely before the slow feature analysis. The parameter k was used to calculate the weight of channel attention. Four strategies of experiments confirmed that the ASFA module has good performance. Furthermore, we used three public standard datasets to train and test in every strategy. A large number of experiments verified the reliability of the ASFA. More importantly, the ASFA model could be applied to practice areas and contribute to the environmental benefit analysis. According to our AMP-CD dataset, we can predict the traffic changes and analyze the economic benefit from traffic expansion.The future work is to extend the adaptive ability of the change detection model and study the fine-grained detection.

**Author Contributions:** Conceptualization, W.J. and S.Z.; methodology, S.Z. and P.K.; software, P.K. and J.W.; validation, J.W., S.C. and G.C.; formal analysis, S.Z., P.K. and S.C. ; investigation, W.J. and S.Z.; resources, W.J., G.C. and H.S.; data curation, P.K. and J.W.; writing—original draft preparation, S.Z.; writing—review and editing, W.J. and S.Z.; visualization, S.C.; supervision, W.J. and G.C.; project administration, W.J. and H.S.; funding acquisition, W.J. and G.C. All authors have read and agreed to the published version of the manuscript.

**Funding:** This paper was funded by the National Natural Science Foundation of China (32171777), by Fundamental Research Funds for the Central Universities (2572020AW48), by Heilongjiang Province Applied Technology Research and Development Program Major Project (GA20A301), by the National Key R & D Plan "Intergovernmental International Scientific and Technological Innovation Cooperation" (Grant Number: 2021YFE0117000), and by the Special Science Fund for the Innovation Ecosystem Construction of the National Supercomputing Center in Zhengzhou (No. 201400210100).

**Data Availability Statement:** The SZTAKI dataset is freely available [34], the SYSU-CD dataset is freely available [35], and the LEVIR-CD dataset is freely available [21].

**Conflicts of Interest:** The authors declare no conflict of interest.

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
