# Peer review of "Remote Sensing Change Detection Based on Unsupervised Multi-Attention Slow Feature Analysis"

_remotesensing, doi:10.3390/rs14122834_

Round 1
Reviewer 1 Report
This paper proposes a new unsupervised method for change detection. The BoVW model was used for feature extraction. The K-means model was used to partition the extracted features into clusters. The ResNet with a multi-attention mechanism (through three dimensions: channels, special, and temporal) was used to generate the feature map. Then, the slow feature analysis was used to generate the change map of the image. This article is well organized and tackles a problem of paramount importance related to traffic monitoring. Thus, the article can be accepted for publication after performing some minor corrections:
- The first sentence in the introduction is not clear: “The transport system industry plays a critical role in the impact on the environment and the achievement of carbon neutrality”.
- The introduction section is too long. Please, try to create a new section named related works, where you can present the methods of change detection.
- It is not clear from the method section what is the type of data used as inputs and targets to the Attention ResNet model.
- Please, try to provide a graph describing your proposed model where the data type flow between the different models is specified (Figure 2 is not clear to follow).
- Please, verify the punctuation and the grammar.
Reviewer 2 Report
The technical quality of the manuscript is reasonable. However, the reviewer found the title inappropriate. The so-called "for transport benefit" is merely about detecting changes in road/bridge construction. Furthermore, the techniques developed in this manuscript are applicable to any change detection problem. In other words, the proposed techniques did not take into account the characteristic features of the transportation traffic. Therefore, the reviewer would like to suggest the authors should consider removing the "for transport benefit" from the title. Finally, there are many careless typos in the manuscript. For instance, Line 181, "falling" should be "following". Also, space " " should be inserted after ".". For instance, Line 186-187. The authors are strongly suggested to revise the whole manuscript carefully.
Round 2
Reviewer 2 Report
The quality of the manuscript has been improved. The reviewer would like to recommend the acceptance of the submission for publication.
